

# Selection and evaluation of lactic acid bacteria from chicken feces in Thailand as potential probiotics

Benjamas Khurajog[1], Yuda Disastra[1], Lum Dau Lawwyne[1], Wandee Sirichokchatchawan[2,3], Waree Niyomtham[1,3], Jitrapa Yindee[1,3], David John Hampson[4] and Nuvee Prapasarakul[1,3]

[1] Department of Veterinary Microbiology, Faculty of Veterinary Science., Chulalongkorn University, Bangkok, Thailand
[2] College of Public Health Sciences, Chulalongkorn University, Bangkok, Thailand
[3] Center of Excellence in Diagnosis and Monitoring of Animal Pathogens (DMAP), Chulalongkorn University, Bangkok, Thailand
[4] School of Veterinary Medicine, Murdoch University, Perth, Western Australia, Australia

Corresponding author
Nuvee Prapasarakul,
Nuvee.P@chula.ac.th

## ABSTRACT

**Background:** Lactic acid bacteria (LAB) are widely used as probiotics in poultry production due to their resilience to low pH and high bile salt concentrations, as well as their beneficial effects on growth performance and antagonistic activity against enteric pathogens. However, the efficacy of probiotics depends on strain selection and their ability to colonize the host's intestine. This study aimed to select, identify, and evaluate LAB strains isolated from chicken feces in Thailand for potential use as probiotics in the chicken industry.

**Methods:** LAB strains were isolated from 58 pooled fresh fecal samples collected from chicken farms in various regions of Thailand, including commercial and backyard farms. Gram-positive rods or cocci with catalase-negative characteristics from colonies showing a clear zone on MRS agar supplemented with 0.5% $CaCO_3$ were identified using MALDI-TOF mass spectrometry. The LAB isolates were evaluated for acid (pH 2.5 and pH 4.5) and bile salt (0.3% and 0.7%) tolerance. Additionally, their cell surface properties, resistance to phenol, antimicrobial activity, hemolytic activity, and presence of antimicrobial resistance genes were determined.

**Results:** A total of 91 LAB isolates belonging to the *Pediococcus, Ligilactobacillus, Limosilactobacillus*, and *Lactobacillus* genera were obtained from chicken feces samples. Backyard farm feces exhibited a greater LAB diversity compared to commercial chickens. Five strains, including *Ligilactobacillus salivarius* BF12 and *Pediococcus acidilactici* BF9, BF14, BYF20, and BYF26, were selected based on their high tolerance to acid, bile salts, and phenol. *L. salivarius* BF12 and *P. acidilactici* BF14 demonstrated strong adhesion ability. The five LAB isolates exhibited significant cell-cell interactions (auto-aggregation) and co-aggregation with *Salmonella*. All five LAB isolates showed varying degrees of antimicrobial activity against *Salmonella* strains, with *P. acidilactici* BYF20 displaying the highest activity. None of the LAB isolates exhibited beta-hemolytic activity. Whole genome analysis showed that *L. salivarius* BF12 contained *ermC, tetL*, and *tetM*, whereas *P. acidilactici* strains BF9 and BF14 carried *ermB, lnuA*, and *tetM*.

**Conclusion:** The selected LAB isolates exhibited basic probiotic characteristics, although some limitations were observed in terms of adhesion ability and the

presence of antibiotic resistance genes, requiring further investigation into their genetic location. Future studies will focus on developing a probiotic prototype encapsulation for application in the chicken industry, followed by *in vivo* evaluations of probiotic efficacy.

# INTRODUCTION

The intensification of poultry production and the rising demand for poultry meat have led to increased antimicrobial drug usage in poultry farms. Subtherapeutic doses of antimicrobial drugs are often administered for prophylactic purposes or as growth promoters; however, this practice raises concerns as it can contribute to the emergence and dissemination of antimicrobial-resistant pathogens (*de Mesquita Souza Saraiva et al., 2022*). To address this issue, alternatives to antibiotic use in food-producing animals, such as probiotics, have been developed (*Gadde et al., 2017*).

Probiotics, as defined by the World Health Organization, are live microorganisms that confer health benefits to their hosts (*FAO/WHO, 2001*). Probiotics increasingly have gained recognition as an alternative to antibiotics in animal production, including poultry. Numerous studies have reported the advantages of probiotic supplementation in poultry, such as enhanced growth performance, improved feed efficiency (*Khatun et al., 2022*; *Reuben et al., 2022*), modulation of cecal microflora composition (*Qiu et al., 2022*), and reduction of *Salmonella* colonization in the gastrointestinal tract (*Khan & Chousalkar, 2020*; *Khochamit et al., 2020*). However, it is crucial to note that the effects of probiotics are strain-dependent (*Butel, 2014*).

The utilization of lactic acid bacteria (LAB) as probiotics in poultry production has gained significant attention due to their ability to thrive in harsh gastrointestinal conditions and confer beneficial effects such as enhanced growth performance and antagonism against enteric pathogens (*El-Sawah et al., 2020*). However, the fact that the efficacy of probiotics is strain-specific emphasizes the importance of selecting appropriate strains for specific host-origin applications (*Kalia et al., 2017*). Currently, there is a critical knowledge gap regarding the identification, characterization, and evaluation of host-associated LAB strains isolated from chicken feces in Thailand for their potential use as probiotics in the local chicken industry.

Chick hatching and post-hatch fasting are significant aspects of poultry practices that can lead to colonization of the intestinal tract by pathogenic bacteria. During this critical period, newly hatched chicks are particularly vulnerable to infections, with exposure to potential pathogens significantly impacting the maturation of their gut microbiome and immune system, consequently exerting influence on their overall well-being and growth performance (*Marcolla, Alvarado & Willing, 2019*; *Siwek et al., 2018*). To mitigate the colonization of harmful bacteria, it is advisable to administer probiotics within the first week after hatching. This strategic supplementation enhances the likelihood of successful

colonization within the developing intestinal tract (*de Oliveira et al., 2014*). To optimize colonization, it is recommended to provide a suitable dose of probiotics, ensuring a minimum viable cell concentration of $1 \times 10^6$ colony forming units (CFU) per g of supplement (*Ramlucken et al., 2021*). For increased efficacy and a higher abundance of beneficial bacteria, supplementation levels can be escalated to a range of $1 \times 10^8$ to $1 \times 10^9$ CFU/g (*Patterson & Burkholder, 2003*). Several studies have reported that administering probiotics through drinking water is more efficient compared to supplementing them in feed (*Blajman et al., 2014*; *Eckert et al., 2010*). This superiority can be attributed to the expedited transit of probiotics through the upper gastrointestinal tract when delivered *via* drinking water, thereby minimizing exposure to acidic pH levels and bile salts (*Karimi Torshizi et al., 2010*).

The current study included examining the effects of selected probiotic strains on *Salmonella* serovars, since Salmonellosis poses a significant challenge to the poultry industry, impacting both economic factors and animal health. Infected chickens experience reduced growth rates, decreased productivity, and increased morbidity and mortality rates, particularly in young chickens (*Gast & Porter, 2020*). Even in infected older chickens that do not exhibit clinical signs, they may serve as carriers and shed *Salmonella* bacteria in their feces, potentially contributing to the dissemination of *Salmonella* contamination during poultry meat processing (*Antunes et al., 2016*).

Lactic acid bacteria (LAB), including species from the genera *Lactobacillus*, *Streptococcus*, *Pediococcus*, *Enterococcus*, and *Weissella*, are commonly used as bacterial probiotics in poultry production (*Hernandez-Patlan et al., 2020*). The selection of probiotics should be based on strains derived from the target host species, as they have a higher likelihood of survival and colonization within the gastrointestinal tract, thereby providing optimal benefits to the host. Furthermore, probiotics must meet the safety criteria outlined by the European Food Safety Authority, which includes the absence of acquired antimicrobial resistance (*Rychen et al., 2018*; *Gopal & Dhanasekaran, 2021*). Key requirements for probiotic properties include viability in acidic conditions, tolerance to bile acids, adherence and colonization in the intestinal epithelium, antagonistic activity against pathogenic bacteria, and, to ensure safety, assessment to avoid strains exhibiting hemolytic activity and/or the presence of antimicrobial resistance genes (*Markowiak & Śliżewska, 2018*).

The current study aimed to address gaps in knowledge by selecting, identifying, and evaluating host-associated LAB isolated from chicken feces in Thailand for their potential use as probiotics in the chicken industry. Functional and safety aspects of these isolates were assessed to determine their suitability as probiotics. By considering strain-specific properties, such as acid and bile tolerance, adhesion ability, antagonism against pathogens, and absence of safety concerns, this research contributes to the development of effective probiotics specifically tailored for the poultry industry.

## MATERIALS AND METHODS

### Sample collections

Swabs of freshly passed feces were used to prepare 58 pooled fecal samples that originated from eight commercial farms (28 pooled feces samples from laying hens and 24 pooled samples from broiler chickens) and six pooled fresh samples from 27 Thai-native chickens from three household farms. The age groups of the sampled chickens were 37–40 weeks for laying hens, 5–6 weeks for broilers, and 12–16 weeks for Thai-native chickens. The rearing system on the commercial farms included cages for layers and open pens for broilers, commercial feeding, and sub-therapeutic doses of antibiotics for broilers to prevent infectious disease until 19 days of age. On the other hand, the rearing system on the backyard farms included free-range husbandry around the house, feeding grains and natural foraging, and no use of antibiotics.

### Lactic acid bacteria isolation

Acidified MRS (de Man, Rogosa and Sharpe) broth, which contained 0.02% (w/v) sodium azide was adjusted to a pH of 5.5 using 5 NHCl, and then was used to store the fecal samples while preventing the growth of Gram-negative bacteria. The samples were kept at 4 °C during transportation to the laboratory and then were incubated at 37 °C for 24–48 h (*Ahn et al., 2002*). Bacterial cultures in MRS broth were streaked in triplicate on MRS agar containing 0.5% $CaCO_3$. After 48 h of incubation at 37 °C, colonies with a clear zone were identified as lactic acid bacteria. The colonies were chosen at random, purified on MRS agar, and validated using Gram staining and the catalase test. For further examination, only Gram-positive, catalase-negative isolates were chosen. The LAB isolates were kept at −20 °C in MRS broth supplemented with 20% (w/v) glycerol. This experiment was approved by the Faculty of Veterinary Science Institutional Biosafety Committee (agreement no. IBC20310148). All selected lactic acid bacterial strains were submitted and maintained in liquid nitrogen tanks at the Pathogen Bank of the Faculty of Veterinary Science, Chulalongkorn University.

### Identification of lactic acid bacteria

Matrix-assisted laser desorption/ionization-time-of-flight mass spectrometer (MALDI-TOF) (Bruker, Mannheim, Germany) was used for genus and species identification. Single bacterial colonies were spotted onto MALDI target plates. Subsequently, the bacterial sample was overlaid with 1 μl of 70% formic acid and then with 1 μl matrix solution containing 10 mg/mL HCCA (a-cyano-4-hydroxycinnamic acid, Sigma-Aldrich, Poland) dissolved in 50% acetonitrile (Sigma-Aldrich, Poznań, Poland) and 2.5% TFA (trifluoro-acetic acid, Sigma-Aldrich, Poland), and air-dried at room temperature. The target plate was loaded into the spectrometer for automated measurement and data interpretation. The mass spectra were processed with the MALDI Biotyper 3.0 software package (Bruker, Germany). The results were shown as the top 10 identification matches. According to the criteria recommended by the manufacturer, a log (score) below 1.70 does not allow for reliable identification; a log (score) between 1.70 and 1.99 allows identification to the genus level; a log (score) of up to 2.00 indicates highly probable identification at species level (*Dec et al., 2016*).

## Evaluation of probiotic functional properties of LAB isolates
### Survival of LAB at low pH and varying bile salt concentrations
*Pre-screening for resistance at pH 2.5 and with 0.3% bile salts*

A total of 91 LAB isolates were evaluated in MRS broth acidified to pH of 2.5 with 1M HCL and MRS broth supplemented with bile salts (0.3% (w/v) Oxgall powder (Sigma-Aldrich, St. Louis, MO, USA)). Briefly, the concentration of an overnight LAB culture was adjusted to $10^8$ CFU/ml (OD600 = 1.0) with MRS broth, and two test tubes were inoculated (5 ml per tube). The bacterial cells were extracted by centrifuging at 4,000 g for 3 min, after which the supernatant was discarded. Either 5 ml of acidified MRS broth or MRS broth enriched with 0.3% (w/v) Oxgall powder was used to resuspend the pellets (Sigma-Aldrich, St. Louis, MO, US). The tubes were incubated for 3 h at 37 °C. Counts of viable cells were conducted using the drop plate method. Briefly, the bacterial solution was serially diluted with 0.85% NaCl, MRS agar was spot inoculated with six drops of 10 μl, and then incubated at 37 °C for 24 h. Between 3 and 30 colonies were chosen to determine the number of viable cells (*Herigstad, Hamilton & Heersink, 2001*). Twenty LAB isolates had at least $10^4$ CFU/ml of live cells and these were chosen for the further testing as described in the following sections.

*Resistance to acid at various pH levels and bile salt concentrations at a temperature of 42 °C*

The resistance test was conducted as described by *Feng et al. (2017)*, with minor modifications. To determine the survival of LAB in the simulated gastrointestinal tract of chickens, the selected LAB isolates were tested for their resistance to different pH values and bile concentrations at a temperature of 42 °C: they were agitated in acidified MRS broth (pH 2.5 and pH 4.5; incubated for 3 and 12 h, respectively) and MRS broth supplemented with bile salts (0.3% and 0.7%; incubated for 6 h). Viable cells in MRS broth without acidification (pH 6.2) served as a control. The viable cell count was conducted as described previously and the tests were performed in duplicate. The survival rate (% Resistance) was calculated as ($\log_{10}$ of viable cell counts in MRS broth in different pH or bile concentrations/$\log_{10}$ of viable cell counts in non-acidified MRS broth) × 100.

### Resistance to 0.4% phenol
The approach described by *Hossain et al. (2021)* was implemented with minor modifications. Five LAB isolates that exhibited high tolerance to different pH levels and concentrations of bile salts at 42 °C were adjusted to $10^8$ CFU/ml and 200 μl aliquots were inoculated into 10 ml MRS broth containing 0.4% phenol and incubated at 37 °C. Viable cell counts (CFU/ml) were determined in triplicate at times 0 and 24 h. The percentage survival rate was calculated as ($\log_{10}$ of viable cell counts at 24 h/$\log_{10}$ of viable cell counts at 0 h) × 100.

### Cell surface properties
*Cell surface hydrophobicity*

The five selected LAB isolates were grown for 24 h. Following centrifugation at 6,000 g for 15 min at 4 °C, bacterial cells were collected, washed twice with PBS, and resuspended in PBS to an optical density (OD) of 0.6 at 600 nm. One ml of either xylene or toluene was

added to different tubes containing 3 ml of bacterial suspensions, which were then vortexed for 90 sec and incubated at room temperature for 30 min. The lower aqueous phase was collected and the OD at 600 nm was determined. The tests were performed in triplicate. The surface hydrophobicity % was calculated as $[(OD_{600}$ before mixing $- OD_{600}$ after mixing$)/(OD_{600}$ before mixing$)] \times 100$ (*Ekmekci, Aslim & Ozturk, 2009*).

*Auto-aggregation*
The chosen LAB isolates were grown for 24 h. The bacterial cells were then harvested by centrifugation at 6,000 g for 15 min at 4 °C, twice washed with PBS, and resuspended in PBS to an OD of 0.6 at 600 nm ($A_{0\,h}$). Three ml of each bacterial suspension was aliquoted into four tubes, vortexed for 10 s and incubated at 37 °C. The absorbance of supernatant at 600 nm ($A_{final\,h}$) was measured at 1, 2, 3, and 4 h. The test was performed in triplicate. Auto-aggregation (%) was calculated as $(1 - A_{final\,h}/A_{0\,h}) \times 100$ (*Xu et al., 2009*).

*Co-aggregation*
The selected LAB cultures in MRS broth and 23 isolates from chickens belonging to nine serovars of *Salmonella enterica*, included seven strains of Typhimurium, six strains of Enteritidis, and two strains each of Agona, Virchow, Kentucky, Hadar, Albany, Braenderup and Give were cultured in Nutrient broth, harvested by centrifugation at 6,000 g for 15 min at 4 °C, washed twice, resuspended with sterile PBS and adjusted to an OD of 0.6 at 600 nm. Equal volumes (2 ml) of the LAB isolate and the pathogenic strain were mixed for 10 s and incubated at 37 °C for 4 h. The absorbance was then measured at 600 nm ($OD_{mix}$). The test was performed in triplicate. Co-aggregation (%) was calculated as $100 \times [(OD_{LAB} + OD_{pathogen}) - 2(OD_{mix})]/(OD_{LAB} + OD_{pathogen})$ (*Ekmekci, Aslim & Ozturk, 2009*).

### Antimicrobial activity against Salmonella enterica
The cell-free supernatants (CFS) were produced from the chosen LAB isolates to assess antibacterial activity against *Salmonella* using a modified agar well diffusion test (*Lin et al., 2006*). Briefly, five LAB isolates were inoculated at $10^8$ CFU/ml in 30 ml MRS liquid medium and cultivated at 37 °C for 24 h. CFS were obtained by centrifugation at 7,000 rpm at 4 °C for 5 min before passing through a 0.22 sterile filter (Millipore, Bedford, Massachusetts). Neutralized cell-free supernatant (NCFS) was prepared by adjusting the pH of aliquoted CFS with 1N NaOH to 6.5–7. The nine serovars of *Salmonella enterica* were employed as pathogenic indicator bacteria. The *Salmonella* serovars were grown in nutrient broth at 37 °C for 18 h, adjusted to a concentration of $10^8$ CFU/ml with Nutrient broth, and 100 µl of the culture was spread onto nutrient agar plates. Wells of 8 mm diameter were cut into the agar and 100 µl of CFS, NCFS, and MRS broth, which served as a negative control, were added. All plates were incubated for 24 h at 37 °C. After incubation, the results were represented in millimeters of zone diameter of inhibition (ZDI) values and interpreted as less active (+), moderately active (++), strongly active (+++), and very strongly active (++++) for ZDIs of 10, 11–14, 15–19, and 20 mm, respectively. Three independent experiments were conducted.

## Evaluation of probiotic safety
### Antimicrobial resistance detection
*Antimicrobial susceptibility*

The antimicrobial susceptibilities of the five selected LAB isolates were determined using the disc diffusion method on MRS agar modified from the method described by the Clinical and Laboratory Standards Institute (*CLSI, 2012*). Antibiotic discs (BD, Sparks, MD, USA) consisting of ampicillin (10 µg), gentamicin (10 µg), kanamycin (30 µg), streptomycin (10 µg), erythromycin (15 µg), clindamycin (2 µg), tetracycline (30 µg), and chloramphenicol (30 µg), were used for the susceptibility determination. Inhibition zone diameters were interpreted according to *Charteris et al. (1998)*.

*Genotyping antimicrobial resistance assay using a real-time PCR with specific probes*

DNA was extracted from the five selected LAB isolates utilizing Genomic DNA NucleoSpin® Tissue (Macherey-Nagel). The genes encoding resistance to antimicrobial classes including penicillin, amoxicillin, cephalosporins, carbapenems, folate pathway inhibitors, polymyxins, tetracyclines, phenicols, aminoglycosides, macrolides, and quinolones, as listed in Data S1, were detected using a genotypic antimicrobial resistance assay as outlined in a previous study (*Pholwat et al., 2019*). In brief, primer/probe sets at final concentrations of 0.9 and 0.25 µM, respectively, were combined with a 5 µl PCR mixture containing 2.5 µl of 2x PCR buffer, 0.2 µl of 25x PCR enzyme from the AgPath-ID-PCR kit (Applied Biosystems, Life Technologies Corporation, Waltham, MA, USA), 0.89 µl of nuclease-free water, and 1 µl of DNA sample. The combination was loaded into 384 well plates on the ViiA7 instrument (Applied Biosystems, Life Technologies Corporation, Waltham, MA, USA). The assay reaction was run with an initial denaturation at 95 °C for 10 min, followed by 40 cycles of denaturation at 95 °C for 15 s and annealing/extension at 60 °C for 1 min. Either well-characterized bacterial isolates or synthetic fragment/plasmid controls (Genewiz Inc., South Plainfield, NJ, USA) were used as positive controls. The genomic DNA of *E. coli* ATCC 25922 and nuclease-free water was used as a negative control.

*Whole-genome search for antimicrobial resistance genes*

DNA was extracted from the five selected LAB isolates using the ZymoBIOMICS™ DNA Miniprep kit (Zymo Research Corp., Irvine, CA, USA) and submitted to Novogene Bioinformatics Technology Co. Ltd., Beijing, China to performed short-read sequencing on the Illumina NovaSeq-PE150 platform with 1 GB data output. The paired-end raw sequence reads were filtered out to remove low-quality sequences with scores <30 using Trimmomatic v.0.38 (*Bolger, Lohse & Usadel, 2014*) and the genome assemblies were created using unicycler v.0.5.0 (*Wick et al., 2017*). All genomes are available in the NCBI genome under the BioProject accession number PRJNA1025932. The antimicrobial resistance genes were analyzed by Staramr v.0.10.0 with default parameters: 98% identity for BLAST, 60% length overlap for BLAST hit in the ResFinder database, 95% length overlap for BLAST hit in the PointFinder database, and 60% length overlap for BLAST hit on the PlasmidFinder database (*Bharat et al., 2022*).

*Hemolytic activity*

Overnight cultures of LAB in MRS broth were streaked on tryptic soy agar (Sigma-Aldrich, Munich, Germany) supplemented with 5% (w/v) sheep blood. After 24 h incubation at 37 °C, the plates were examined for hemolysis. *S. aureus* ATCC 25923 was used as the positive control. Hemolytic activities around the colonies were recorded as follows: Beta (β) hemolysis was a clear, colorless/lightened yellow zone; Alpha (α) hemolysis was a small zone of greenish to brownish discoloration of the media; and Gamma (υ) hemolysis was no change observed in the media–recorded as non-hemolytic (*Argyri et al., 2013*; *Lee et al., 2011*).

## Statistical analysis

The parameters in this study were presented as mean ± SD. Probiotic properties were compared among LAB isolates using one-way ANOVA, except the resistance at pH 4.5 was compared using the independent-samples Kruskal-Wallis test. Paired t-tests were used to compare resistance to bile salts (0.3% and 0.7%), and hydrophobicity (toluene and xylene). Repeated measurement ANOVA was used to compare auto-aggregation (1, 2, 3 and 4 h). Spearman's correlation coefficient was used to establish a relationship between hydrophobicity, auto-aggregation, and co-aggregation. Significant difference was set at $p < 0.05$. All the statistical analyses were performed using IBM SPSS Statistics (Version 28) and GraphPad Prism version 9 for macOS (GraphPad Software Inc., La Jolla, CA, USA).

# RESULTS

## Isolation and identification of lactic acid bacteria

A total of 91 presumptive LAB isolates were obtained, displaying clear zones around the colonies on MRS agar supplemented with 0.5% $CaCO_3$. These Gram-positive bacilli or cocci were further identified as LAB species using MALDI-TOF MS. The distribution of LAB species varied among different chicken types, as depicted in Fig. 1A and Table S1. The LAB isolates were identified as belonging to six species in four genera: *Pediococcus, Ligilactobacillus, Limosilactobacillus*, and *Lactobacillus*. Five different species were found in Thai-native chickens from backyard farms, compared to four in laying hens and two in broilers. *Pediococcus acidilactici* was the most common species found in broilers and Thai-native chickens, while *L. salivarius* and *P. pentosaceus* were the most common species detected in laying hens.

## Survival of LAB under low pH and at different concentrations of bile

The pre-screening of resistance at pH 2.5 and 0.3% bile identified 20 LAB isolates that grew under both these conditions, with viable cell counts ranging from $10^4$ to $10^7$ CFU/ml. However, only five LAB isolates exhibited high tolerance to different pH levels and concentrations of bile salts at 42 °C. Three of these specific isolates were derived from one farm raising broiler chickens (*L. salivarius* BF12 and *P. acidilactici* BF9 and BF14), while the other two were from a farm with Thai-native chickens (*P. acidilactici* BYF20, and BYF26) (Fig. 1B, Tables 1 and S2). These five isolates were used in all subsequent testing. *P. acidilactici* BF9 demonstrated the highest survival rate in acidified MRS (pH 2.5 and pH

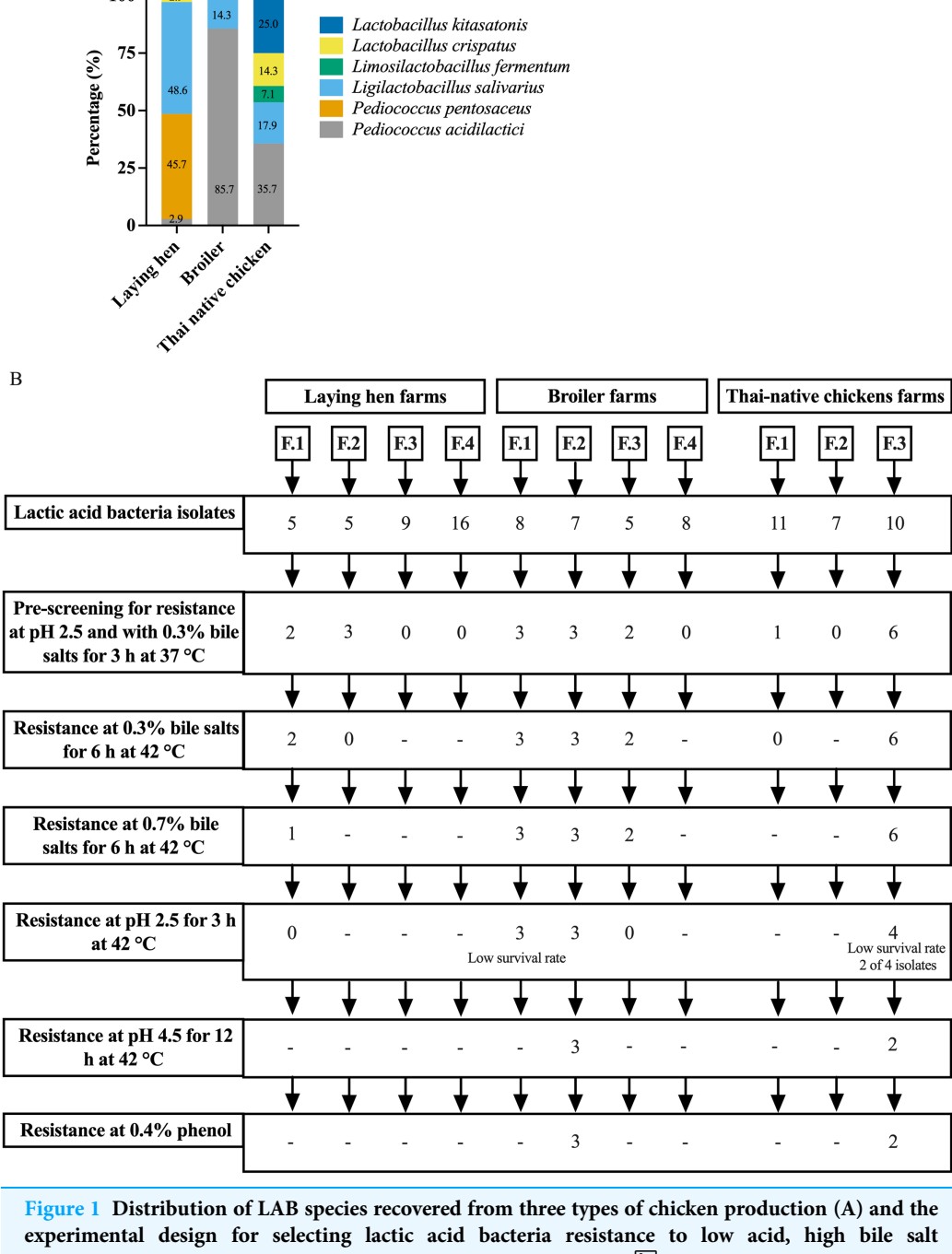

**Figure 1** Distribution of LAB species recovered from three types of chicken production (A) and the experimental design for selecting lactic acid bacteria resistance to low acid, high bile salt concentrations and 0.4% phenol (B).

4.5) and 0.3% bile salts, with rates of 72.62%, 98.93%, and 84.77%, respectively, compared to all the selected LAB ($p < 0.05$). Nevertheless, the survival rates of the selected LAB isolates were affected by increasing concentrations of bile salts. The viability of LAB declined when exposed to 0.7% bile salts, although *P. acidilactici* (BF9, BYF20 and BYF26) exhibited higher survival rates in 0.7% bile salts ($p < 0.05$), and particularly *P. acidilactici* BYF26 which displayed the highest survival rate at 75.54%.

**Table 1 Survival of LAB in various conditions.**

| LAB isolate | (%) Resistance | | | | |
| --- | --- | --- | --- | --- | --- |
| | Acid | | Bile salts (6 h) | | Phenol (24 h) |
| | pH 2.5 (3 h) | pH 4.5 (12 h) | 0.3% | 0.7% | 0.4% |
| *L. salivarius* BF12 | 58.58 ± 1.22[bc] | 98.42 ± 1.18[a] | 71.75 ± 1.63[b, *] | 49.08 ± 0.38[b, **] | 132.79 ± 2.54[a] |
| *P. acidilactici* BF9 | 72.62 ± 0.48[a] | 98.93 ± 0.25[a] | 84.77 ± 1.33[a, *] | 68.56 ± 0.78[a, **] | 103.97 ± 10.53[ab] |
| *P. acidilactici* BF14 | 56.58 ± 0.13[c] | 96.99 ± 0.66[a] | 72.20 ± 0.30[abc, *] | 47.25 ± 4.50[ab, **] | 114.95 ± 20.22[ab] |
| *P. acidilactici* BYF20 | 52.76 ± 0.04[b] | 78.09 ± 0.01[a] | 78.16 ± 0.54[ab, *] | 72.11 ± 0.57[a, **] | 102.93 ± 5.96[b] |
| *P. acidilactici* BYF26 | 67.10 ± 2.00[abc] | 98.37 ± 0.70[a] | 82.11 ± 0.81[ab, *] | 75.54 ± 1.38[a, **] | 114.07 ± 3.76[b] |

Notes:
Survival of LAB under low pH conditions, different concentrations of bile salts and with phenol (0.4%) following incubation at 42 °C.
[a, b, c] Within a column indicates significant differences between LAB isolates ($p < 0.05$).
[*, **] Within a row indicates significant differences ($p < 0.05$) when compared with the resistance percentages for bile salts at 0.3% and 0.7%.

## Phenol tolerance

All five selected LAB isolates showed great resistance toward 0.4% phenol, with values ranging from 102.93% to 132.79%, indicating growth in the presence of phenol (Tables 1 and S2). *L. salivarius* BF12 had the highest viability with a resistance rate of 132.79 % ($p < 0.05$).

## Cell surface properties

The cell surface hydrophobicity calculated for the five selected LAB isolates was not significantly different in either xylene or toluene ($p < 0.05$) (Fig. 2A and Table S3 (A)). The hydrophobicity percentages varied from −1.97% to 94.31% depended on the LAB isolates. *L. salivarius* BF12 exhibited the highest hydrophobicity percentages towards both xylene and toluene compared with all the selected LAB isolates ($p < 0.05$). In contrast, *P. acidilactici* BYF26 had no adherence with xylene or toluene.

All five selected LAB isolates showed a high propensity to undergo auto-aggregation, with increasing auto-aggregation percentages throughout the time periods from 1–4 h (Fig. 2B and Table S3 (B)). At 4 h, the lowest ability (11.13%) was demonstrated by *P. acidilactici* BYF20, whereas *L. salivarius* BF12 had the highest ability at 62.79% when comparing between different LAB isolates and incubation periods ($p < 0.05$).

The results of co-aggregation percentages of the selected LAB isolates with nine serovars of *S. enterica* varied between 6.07% to 31.14% (Fig. 2C and Table S3 (C)). *L. salivarius* BF12 had the highest ability to aggregate with all nine serovars of *S. enterica* compared with the other selected LAB isolates tested with the same *Salmonella* strains ($p < 0.05$).

## Antimicrobial activity against *Salmonella enterica*

CFS from the five selected LAB isolates exhibited moderate to very strong inhibition against the 23 isolates of *S. enterica*, representing nine serovars (Tables 2 and S4). In contrast, none of the NCFS from the five LAB isolates inhibited *Salmonella* growth (Fig. S1).

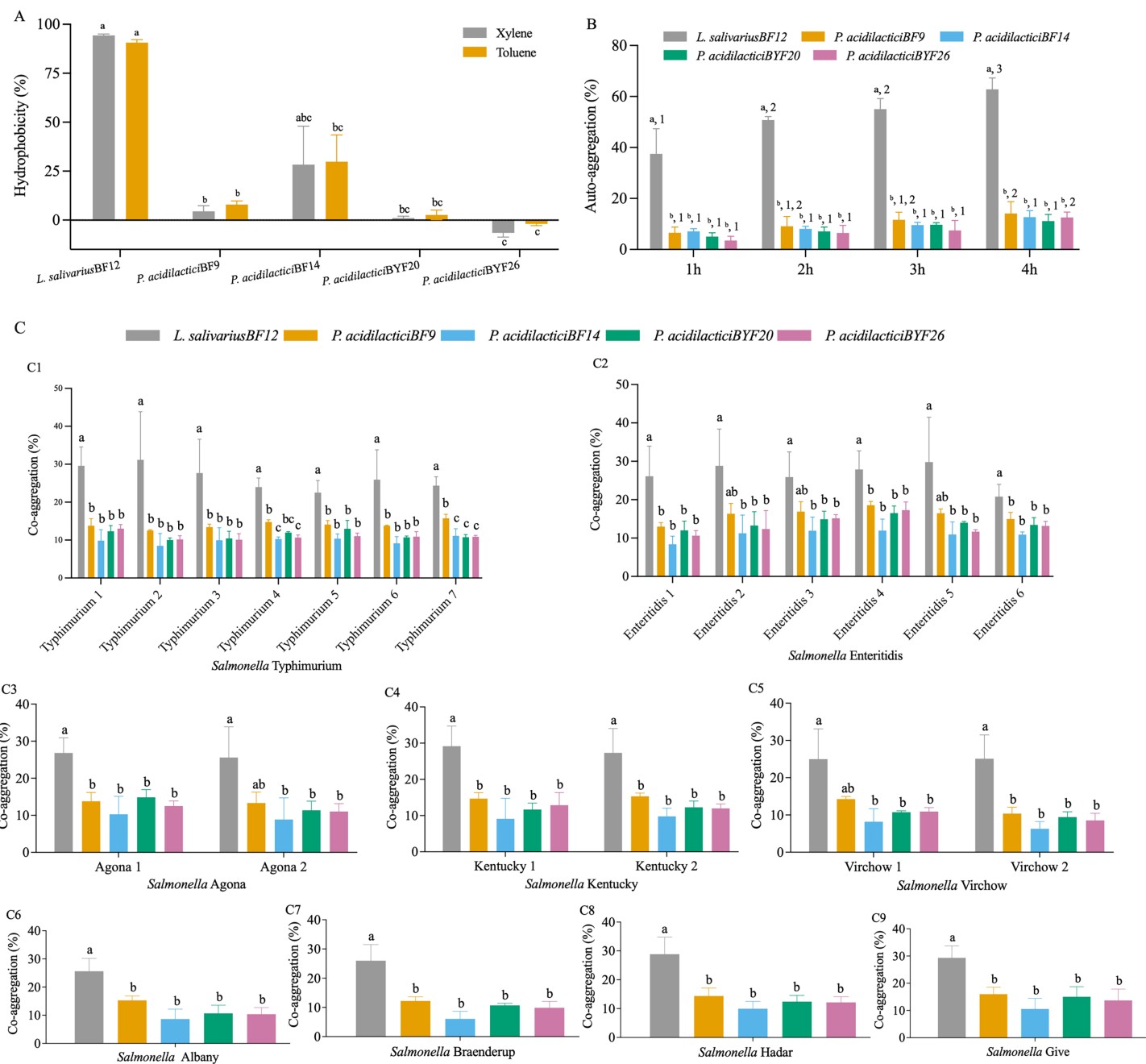

**Figure 2 Cell surface properties of the selected LAB isolates.** (A) Hydrophobicity percentages: the different lowercase letters with the same color bars indicate significant differences between the LAB isolates within the same test ($p < 0.05$). (B) Auto-aggregation percentages: the different lowercase letters at the same time indicate a significant difference ($p < 0.05$) and the different lowercase numbers with the same color bars indicate significant differences between the same isolates at the different times ($p < 0.05$). (C) Co-aggregation percentages: the different lowercase letters indicate a significant difference between the LAB isolates with the similar Salmonella strain ($p < 0.05$).

## Antimicrobial resistance detection

The phenotypic and genotypic antimicrobial resistance profiles of the five LAB isolates are summarized in Table 3, with additional details in Table S5 and Data S2. The disk diffusion assay revealed that all the LAB isolates were susceptible to chloramphenicol, but resistant

**Table 2 Antimicrobial activity of 5 LAB isolates against 23 *Salmonella enterica* strains.**

| Indicator strains | *L. salivarius* BF12 | *P. acidilactici* BF9 | *P. acidilactici* BF14 | *P. acidilactici* BYF20 | *P. acidilactici* BYF26 |
|---|---|---|---|---|---|
| *S.* Typhimurium1 | +++ | +++ | +++ | +++ | +++ |
| *S.* Typhimurium2 | +++ | +++ | +++ | +++ | +++ |
| *S.* Typhimurium3 | +++ | +++ | +++ | +++ | +++ |
| *S.* Typhimurium4 | +++ | +++ | +++ | +++ | +++ |
| *S.* Typhimurium5 | +++ | +++ | +++ | +++ | +++ |
| *S.* Typhimurium6 | +++ | +++ | +++ | +++ | +++ |
| *S.* Typhimurium7 | +++ | +++ | +++ | +++ | +++ |
| *S.* Agona1 | +++ | +++ | +++ | +++ | +++ |
| *S.* Agona2 | +++ | +++ | +++ | +++ | +++ |
| *S.* Kentucky1 | +++ | +++ | +++ | +++ | +++ |
| *S.* Kentucky2 | +++ | +++ | ++++ | ++++ | +++ |
| *S.* Virchow1 | +++ | +++ | +++ | +++ | +++ |
| *S.* Virchow2 | +++ | +++ | +++ | +++ | +++ |
| *S.* Albany | +++ | +++ | +++ | +++ | ++ |
| *S.* Braenderup | +++ | +++ | +++ | +++ | +++ |
| *S.* Hadar | +++ | +++ | +++ | ++++ | +++ |
| *S.* Enteritidis1 | +++ | +++ | +++ | +++ | +++ |
| *S.* Enteritidis2 | ++++ | ++++ | ++++ | ++++ | ++++ |
| *S.* Enteritidis3 | +++ | +++ | +++ | +++ | +++ |
| *S.* Enteritidis4 | +++ | +++ | +++ | +++ | +++ |
| *S.* Enteritidis5 | +++ | +++ | +++ | +++ | +++ |
| *S.* Enteritidis6 | +++ | +++ | +++ | +++ | +++ |
| *S.* Give | +++ | +++ | +++ | +++ | +++ |

Notes:

CFS from the five selected LAB isolates exhibited moderate to very strong inhibition against the 23 isolates of *S. enterica*, representing nine serovars.

The antibacterial activity by CFS was expressed as (+) less active (≤10 mm), (++) moderately active (11–14 mm), (+++) strongly active (15–19 mm) and (++++) very strongly active (≥20 mm)

**Table 3 The phenotypic and genotypic antimicrobial resistance profiles of the five LAB isolates.**

| Bacteria species | Isolates | Genotype | | | | | | | | | |
|---|---|---|---|---|---|---|---|---|---|---|---|
| | | AMP | CN | K | S | E | DA | TE | C | RT- PCR | Genome analysis |
| *L. salivarius* | **BF12** | R | R | R | R | R | R | R | S | – | *ermC, tetL* and *tetM* |
| *P. acidilactici* | **BF9** | S | R | R | R | R | R | R | S | *ermB* | *ermB, lnuA* and *tetM* |
| *P. acidilactici* | **BF14** | S | R | R | R | R | R | R | S | *ermB* | *ermB, lnuA* and *tetM* |
| *P. acidilactici* | **BYF20** | S | R | R | R | S | S | MS | S | – | – |
| *P. acidilactici* | **BYF26** | S | R | R | R | S | S | MS | S | – | – |

Notes:

The disk diffusion assay revealed that all the LAB isolates were susceptible to chloramphenicol, but resistant to gentamicin, kanamycin and streptomycin. Ampicillin susceptible occurred in the LAB isolates, except for *L. salivarius* BF12 which was resistant. *P. acidilactici* isolates BYF20 and BYF26 were susceptible to erythromycin and clindamycin and moderately susceptible to tetracycline, while the other three isolates were resistance to these drugs. However, use of the real-time PCR assay only identified *ermB*, an erythromycin resistance gene, and only in *P. acidilactici* isolates BF9 and BF14. In comparison, genome analysis confirmed that *P. acidilactici* isolates BF9 and BF14 contained *ermB* and *lnuA* (lincomycin) and *tetM* (tetracycline) resistance genes, and *L. salivarius* BF12 contained *ermC* (erythromycin), *tetL* and *tetM* (tetracycline) resistance genes.

R = resistant, S = susceptible and MS = moderately susceptible for: AMP = Ampicillin 10 μg, CN = Gentamicin 10 μg, K = Kanamycin 30 μg, S = Streptomycin 10 μg, E = Erythromycin 15 μg, DA = Clindamycin 2 μg, TE = Tetracycline 30 μg, C = Chloramphenicol 30 μg.

to gentamicin, kanamycin and streptomycin. Ampicillin susceptible occurred in the LAB isolates, except for *L. salivarius* BF12 which was resistant. *P. acidilactici* isolates BYF20 and BYF26 were susceptible to erythromycin and clindamycin and moderately susceptible to tetracycline, while the other three isolates were resistance to these drugs. However, use of the real-time PCR assay only identified *ermB*, an erythromycin resistance gene, and only in *P. acidilactici* isolates BF9 and BF14. In comparison, genome analysis confirmed that *P. acidilactici* isolates BF9 and BF14 contained *ermB* and *lnuA* (lincomycin) and *tetM* (tetracycline) resistance genes, and *L. salivarius* BF12 contained *ermC* (erythromycin), *tetL* and *tetM* (tetracycline) resistance genes. These three isolates were obtained from broiler chickens. No antimicrobial resistance genes were detected in *P. acidilactici* isolates BYF20 and BYF26 obtained from Thai-native chickens.

## Hemolytic activity

The selected LAB isolates did not show beta-hemolytic activity, but *L. salivarius* BF12 exhibited partial hemolysis (alpha-hemolysis).

## DISCUSSION

The approach taken in this study was in line with previous research that emphasized the importance of selecting probiotic strains with high tolerance to low pH values and bile salts. The ability of probiotics to survive and establish themselves in the gastrointestinal tract is crucial for their effectiveness. The LAB isolates in this study demonstrated remarkable acid and bile tolerance, indicating their potential to withstand the harsh conditions of the chicken gut. This knowledge aligns with previous studies that highlighted the significance of acid and bile tolerance in probiotic strains (*Lin et al., 2007*).

The fecal samples were collected from various types of chickens, including layers, broilers, and Thai-native chickens, at specific ages (37–40, 5–6, and 12–16 weeks, respectively). These age ranges in the different types of chickens are associated with high productivity (*Hocking et al., 2003*; *Kpomasse et al., 2021*; *Wattanachant, 2008*), which suggests that these age groups are suitable for obtaining probiotic strains. Numerous studies have reported a potential correlation between gut microbiota composition and increased production (*Sun, Hou & Yang, 2021*), further supporting the selection of these age ranges for sample collection.

The selection of probiotics derived from a relevant host is also a key consideration in probiotic research. The current study focused on LAB probiotics isolated from the feces of various types of chickens in Thailand. This approach increases the likelihood of obtaining novel local LAB strains that exhibit potential probiotic properties and greater adaptation in the chicken gastrointestinal tract. This concords with other studies emphasizing the benefits of using probiotics derived from the same host species (*Gopal & Dhanasekaran, 2021*). The diversity of LAB species observed among different chicken types, including laying hens, broilers, and backyard chickens, may be attributed to various factors, including the chicken type, age, diet, rearing practices, antibiotic use, geographical location, and environmental stressors (*Feye et al., 2020*; *Hubert et al., 2019*). Backyard chickens exhibited a higher diversity of species and strains, which could be attributed to

their rearing practices, with free-range systems allowing access to diverse foods and an absence of antibiotic use. Similar findings from previous studies have reported increased microbiota diversity and higher numbers of lactobacilli in chickens raised in cage-free or organic farming systems, where diverse diets including grass, vegetables, and soil are available (*Bjerrum et al., 2006*; *Feye et al., 2020*; *Hubert et al., 2019*; *Kers et al., 2018*; *Musikasang et al., 2012*).

The survival of probiotics in the chicken gastrointestinal tract (GIT) is crucial for their effectiveness. The harsh conditions of low pH and high concentrations of bile salts encountered during transit can significantly impact probiotic viability (*Church & Pond, 1974*; *Lin et al., 2007*). In this study, the selected LAB isolates demonstrated high survival rates under low pH conditions (2.5 for 3 h) and high bile salt concentrations (0.7% for 6 h). Additionally, they showed tolerance to phenol, which is produced by commensal bacteria in the GIT and can inhibit LAB growth (*Reuben et al., 2019*). These tolerance traits indicate that a substantial number of the selected LAB probiotics can survive transit through the harsh conditions in the chicken gut, allowing them to reach the lower part of the intestine. Nevertheless, phenol tolerance is another crucial aspect to consider when evaluating LAB strains as potential probiotics. The selected LAB isolates exhibited significant resistance to phenol, with *L. salivarius* BF12 displaying the highest viability. This characteristic suggests that *L. salivarius* BF12 may possess a protective mechanism against phenolic compounds encountered in the environment of the chicken gut (*Reuben et al., 2019*).

The cell surface properties of probiotic strains can influence their interactions with the host and other microorganisms. Hydrophobic interactions between the LAB cell surface and the intestinal mucosa often plays a role in the initial stages of bacteria adhesion and aggregation, which can improve colonization, prolong retention in the gut and help in competitive exclusion by occupying adhesion sites in the intestinal mucosa. Auto-aggregation and co-aggregation are the ability of bacterial cells to adhere to the same species and different species or genera, respectively. The occurrence of a substantial aggregation of LAB cells *via* an auto-aggregation mechanism can improve not only colonization, but also increase resistance to removal by peristaltic action in the gut. The co-aggregation of LAB cells and pathogens might reduce the pathogen's ability to adhere to and colonize the gut lining (*Schachtsiek, Hammes & Hertel, 2004*). In this study, *L. salivarius* BF12 and *P. acidilactici* strain BF14 exhibited substantial hydrophobic interaction, and all selected LAB isolates exhibited high auto-aggregation ability, indicating their propensity to form aggregates, which can facilitate colonization in the GIT and promote their beneficial effects (*de Melo Pereira et al., 2018*). Furthermore, the ability of LAB strains to co-aggregate with pathogenic bacteria such as *Salmonella enterica* is desirable, as it can contribute to the inhibition of pathogen colonization and subsequent infection. The correlation coefficients between hydrophobicity, auto-aggregation and co-aggregation are shown in Table S3 (D). The auto-aggregation and the adhesion in either xylene or toluene (hydrophobicity) were highly correlated with the co-aggregation of the five LAB isolates with *Salmonella* serovars Albany, Braenderup, Give and Hadar. A negative correlation occured between either the auto-aggregation or hydrophobicity and co-aggregation of the LAB isolates with *S.* Typhimurium, Enteritidis, Kentucky, Virchow
and Agona. This might be due to different co-aggregating mechanisms of LAB interaction with some *Salmonella* serovars, consistent with a previous study that reported that LAB use a specific surface protein (Cpf) to interact with some pathogens (*Schachtsiek, Hammes & Hertel, 2004*). LAB strains that possessed both high hydrophobicity and aggregation abilities may be particularly effective as a probiotic, as they might exhibit enhanced adherence to host tissues, better resistance to expulsion, and competitive exclusion of potential pathogens.

The antimicrobial activity of LAB strains against *S. enterica* is of great interest due to the increasing concern over antibiotic resistance. The selected LAB isolates exhibited varying degrees of antimicrobial activity against *S. enterica* serovars, ranging from moderate to strong inhibition. This antimicrobial activity may be attributed to the production of antimicrobial compounds by the LAB strains, which could help reduce *Salmonella* colonization and minimize the risk of poultry-associated Salmonellosis. Probiotics can exhibit anti-pathogenic activities through various mechanisms, such as co-aggregation with pathogenic bacteria, stimulation of the immune system, competition for nutrients, and production of antimicrobial compounds (*de Melo Pereira et al., 2018*). In this study, all selected LAB isolates, particularly *L. salivarius* BF12, showed a significant ability to co-aggregate with *Salmonella*, suggesting their potential for binding with *Salmonella* cells and competitively inhibiting their adherence to gastrointestinal epithelial cells. Moreover, the cell-free supernatants (CFS) of the selected LAB isolates demonstrated strong to moderate inhibition of *Salmonella* growth. The antimicrobial compounds in the CFS were likely to be active under low pH conditions (3.80–4.08). Previous studies have reported that the antagonistic effect of LAB against pathogens disappears at higher pH or after neutralization (*Bajpai et al., 2016*; *Soria & Audisio, 2014*).

The presence of antimicrobial resistance genes in probiotic strains raises concerns about their potential transfer to commensal bacteria, including opportunistic pathogens, thus contributing to increased antibiotic resistance (*EFSA, 2008*). Whole genome analysis showed that three of the selected isolates carried several antimicrobial resistance genes. Furthermore, the presence of antimicrobial resistance genes *ermC, tetL* and *tetM* in *L. salivarius* BF12 and *ermB, lnuA* and *tetM* in *P. acidilactici* strains BF9 and BF14 was predictive of their phenotypic resistance traits (Table S5), with all three isolates showing phenotypic resistances to erythromycin, clindamycin and tetracycline. Other antimicrobial resistance phenotypes, which could not be linked to identification of antimicrobial resistance genes, were not subjected to further investigate (*EFSA Panel on Additives and Products or Substances used in Animal Feed (FEEDAP) et al., 2018*). These genes might not be verified in an accurate locus on the fragmented genome assemblies from short read sequences, and further analysis of complete genome assemblies from combinations between short-read and long-read sequences is necessary to determine whether the antimicrobial resistance genes are located on mobile genetic elements (*Maboni et al., 2022*). The real-time PCR assay only detected *ermB* in two isolates, and its reliance on specific primers and probe sets emphasizes the superiority of whole genome sequencing for identification of resistance genes in studies of this nature. The three LAB isolates with the identified antimicrobial resistance genes originated from a broiler farm that used

antibiotics, and this use may have encouraged its occurrence. No resistance genes were found in the isolates from Thai chickens from three private farms where no antibiotics were used.

Hemolytic ability is a relevant virulence factor that can be presented in pathogenic microorganisms, and which is best avoided in probiotic isolates. Only one of the five LAB isolates showed hemolytic activity, this being the alpha-hemolytic *L. salivarius* BF12. In previous studies, alpha-hemolytic non-enterococcal LAB have been considered to be safe organisms (*Argyri et al., 2013*; *Lee et al., 2011*; *Touret, Oliveira & Semedo-Lemsaddek, 2018*), so *L. salivarius* BF12 also is likely to be safe for use as a probiotic, subject to further *in vivo* studies.

## CONCLUSIONS

Overall, this study highlights the potential of LAB isolates from chicken feces in Thailand as promising probiotic candidates for the poultry industry. This study identified five LAB isolates, including *L. salivarius* BF12 and *P. acidilactici* BF9, BF14, BYF20, and BYF26, that displayed high tolerance to acid, bile salts, and phenol. These isolates exhibited adhesion ability, except for BYF26, and demonstrated strong anti-pathogenic activities through co-aggregation and the production of antimicrobial compounds against *Salmonella*. Further studies on their *in vivo* efficacy and safety are warranted to determine their full potential as probiotics for enhancing poultry health and mitigating the risks associated with *Salmonella* contamination.

## ACKNOWLEDGEMENTS

We are gratefully to the Pathogen Bank, Faculty of Veterinary Science, Chulalongkorn University, Bangkok, Thailand for supporting work undertaken in this area.

### Funding

The present scientific research was financially supported by the 2022-Fundamental Fund, Thailand Science Research and Innovation (TSRI), Chulalongkorn University (FOOD66310012), the Secondary Century Fund (C2F) for Doctoral Scholarship and the 90th Anniversary of Chulalongkorn University Scholarship (Ratchadaphiseksomphot Endowment Fund) and the ThaiFoods Group public company limited, Bangkok, Thailand. The funders had no role in study design, data collection and analysis, decision to publish, or preparation of the manuscript.

### Grant Disclosures

The following grant information was disclosed by the authors:
Thailand Science Research and Innovation (TSRI).
Chulalongkorn University: FOOD66310012.
Ratchadaphiseksomphot Endowment Fund.
ThaiFoods Group Public Company Limited.

## Competing Interests

The authors declare that they have no competing interests.

## Author Contributions

- Benjamas Khurajog conceived and designed the experiments, performed the experiments, analyzed the data, prepared figures and/or tables, authored or reviewed drafts of the article, and approved the final draft.
- Yuda Disastra performed the experiments, analyzed the data, authored or reviewed drafts of the article, and approved the final draft.
- Lum Dau Lawwyne performed the experiments, prepared figures and/or tables, and approved the final draft.
- Wandee Sirichokchatchawan conceived and designed the experiments, performed the experiments, prepared figures and/or tables, authored or reviewed drafts of the article, and approved the final draft.
- Waree Niyomtham performed the experiments, analyzed the data, prepared figures and/or tables, and approved the final draft.
- Jitrapa Yindee performed the experiments, analyzed the data, prepared figures and/or tables, and approved the final draft.
- David John Hampson analyzed the data, authored or reviewed drafts of the article, and approved the final draft.
- Nuvee Prapasarakul conceived and designed the experiments, performed the experiments, analyzed the data, prepared figures and/or tables, authored or reviewed drafts of the article, and approved the final draft.

## Ethics

The following information was supplied relating to ethical approvals (*i.e.*, approving body and any reference numbers):

All isolates were obtained from freshly passed feces at the chicken farms. The Faculty of Veterinary Science Institutional Biosafety Committee approved this experimental work (agreement number IBC20310148).

## Data Availability

All oligonucleotide sequence primers are available in the Supplemental File.

## Supplemental Information

Supplemental information for this article can be found online at http://dx.doi.org/10.7717/peerj.16637#supplemental-information.

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
