# Peer review of "Selection and evaluation of lactic acid bacteria from chicken feces in Thailand as potential probiotics"

_PeerJ, doi:10.7717/peerj.16637_

## Round 0.1 · original submission · Major Revisions

Dear Authors,

Please address all comments as suggested by the reviewers in a point-by-point manner.

Kind regards,
Academic Editor
PeerJ Life & Environment

Reviewer 1 ·

Basic reporting

Review for Khurajog et al “Selection and evaluation of autochthonous lactic acid bacteria from chicken feces in Thailand as potential probiotics”.


This study has screened 91 Lactic acid bacteria from fecal samples of chicken farms in various regions of Thailand. Out of the 91 isolates they identified 5 LAB suitable as probiotics. These 5 LAB have a high tolerance for both acid (low pH) and bile salt up to 0.7%. They have checked the auto aggregation and co-aggregation of all 5 LAB with salmonella strain. They have also determined the cell surface properties of these 5 LAB. The study presented here is interesting and potentially useful for future probiotics development. The Manuscript is very well written.

I have only minor comments

Please discuss the correlation between hydrophobicity and autoaggregation and coaggregation in the discussion.

Line 310- reframe the sentence “The cell surface hydrophobicity for the five selected LAB isolates in xylene compared with toluene were not significantly different”
to
The cell surface hydrophobicity calculated for the five selected LAB isolates was not significantly different in either xylene or toluene.

Line 330- Provide Agarose gel image of PCR showing the ermB gene amplified in P. acidilactici isolates BF9 and BF14.

Experimental design

no comment

Validity of the findings

no comment

Reviewer 2 ·

Basic reporting

The report is overall clear, but I would like to highlight a few points for consideration:

1. I would call the isolates used host-associated but not autochthonous. I do agree that autochthonous strains are hypothetically more likely to be successful as probiotics, provided they have the positive traits of interest. However, I believe that to classify an isolate or strain as autochthonous, more epidemiological and experimental work must be done to look at colonization dynamics, prevalence, etc. Obviously, I am not offering a definition, and my argument can be debated and disagreed upon. I concur. However, I would just be more cautious with a definition here.

2. I would encourage adding a figure 1 with an experimental design workflow to exemplify from the process of isolation from different farms all the ways to phenotyping, etc, including the number of isolates that were filtered in and out for each step in the process.

3. Although the study lacks an in vivo test, at least for L. salivarius BF12, I agree with the hypothesis and in vitro screening (see below for more comments)

Experimental design

Here are my suggestions to improve the quality of this manuscript:

1. Ultimately, there were 5 isolates (strains) that were used for phenotyping. I believe that whole-genome sequencing and pan-genomic analysis is necessary. This would also answer the question about doing an unbiased search for AMR genes. I believe this is crucial, and it would also confirm, if not, refine the species/strain annotation.

2. I strongly suggest an AMR phenotypic assay of broad spectrum for MICs, on those 5 isolates/strains as well.

3. Maybe Figure 2C can become a separate figure with plots by serovar so the data become more visible. Also, I would suggest providing more information (genotypic if possible) on the strains of S. enterica used.

4. Is there a way to show some figures for the antimicrobial activity properties of the strains against S. enterica isolates?

Validity of the findings

The work is important, but I think it could be enhanced in quality with WGS, MICs, and improved data presentation.

Additional comments

I am encouraged by the work, and think it is important.

---

## Round 0.2 · accepted · Accept

All the comments suggested by the reviewers have been addressed. The manuscript is now accepted.

Reviewer 1 ·

Basic reporting

All my concerns are addressed. I recommend publication.

Experimental design

no comment

Validity of the findings

no comment

Additional comments

no comment

Reviewer 2 ·

Basic reporting

No comments

Experimental design

No comments

Validity of the findings

Improved with the new analysis added

Additional comments

None